# Critical Steps and Common Mistakes during Temporal Bone Dissection: A Survey among Residents and a Step-by-Step Guide Analysis

**DOI:** 10.3390/jpm14040349

**Published:** 2024-03-27

**Authors:** Giovanni Motta, Eva Aurora Massimilla, Salvatore Allosso, Massimo Mesolella, Pietro De Luca, Domenico Testa, Gaetano Motta

**Affiliations:** 1ENT Unit-Department of Mental, Physical Health and Preventive Medicine, University of Campania “Luigi Vanvitelli”, 80121 Naples, Italy; evaaurora.massimilla@unicampania.it (E.A.M.); domenico.testa@unicampania.it (D.T.); gae.motta@libero.it (G.M.); 2Otorhinolaryngology-Head and Neck Surgery Unit, Department of Neuroscience, Reproductive and Odontostomatological Sciences, University of Naples Federico II, 80138 Naples, Italy; salvatore.allosso@unina.it (S.A.); massimo.mesolella@unina.it (M.M.); 3Head and Neck Department, Isola Tiberina-Gemelli Isola Hospital, 00186 Rome, Italy; pietro.deluca.fw@fbf-isola.it

**Keywords:** temporal bone dissection, cortical mastoidectomy (CM), facial recess, facial nerve (FN), posterior tympanotomy (PT), training in oto-surgery, cochlear implantation (CI), cholesteatoma, middle ear surgery

## Abstract

Background: Given that the temporal bone is one of the most complex regions of the human body, cadaveric dissection of this anatomical area represents the first necessary step for the learning and training of the young oto-surgeon in order to perform middle ear surgery, which includes the management of inflammatory pathology, hearing rehabilitation, and also cognitive decline prevention surgery. The primary objective of this study was to identify common mistakes and critical passages during the initial steps of temporal bone dissection, specifically cortical mastoidectomy and posterior tympanotomy. Methods: A survey among 100 ENT residents was conducted, gathering insights into the most prevalent errors encountered during their training to uncover the most challenging aspects faced by novice surgeons during these procedures. Results: The most common mistakes included opening the dura of the middle cranial fossa (MCF), injury of the sigmoid sinus (SS), chorda tympani (CT), and facial nerve (FN) injury while performing the posterior tympanotomy. The most important critical steps to prevent mistakes are related to the absence of wide exposure during cortical mastoidectomy and the consequent impossibility of identifying the landmarks of the facial recess before performing posterior tympanotomy. Injury of these structures was more common in younger surgeons and in the ones who performed less than five temporal bone dissection courses. Conclusions: Numerous temporal bone dissections on cadavers are mandatory for ENT residents looking forward to performing middle ear surgery.

## 1. Introduction

Cortical mastoidectomy (CM) and posterior tympanotomy (PT) are the two most fundamental steps to perform ear surgery, being the two procedures required in the vast majority of middle ear operations, including those for inflammatory pathologies, cochlear implantation (CI), and tumors arising in the middle ear cavity. These procedures are essential for providing surgeons with adequate access to the middle ear and its delicate structures, enabling them to effectively manage and treat a wide range of ear conditions [1,2,3,4]. To be successful in this type of intervention, cadaveric dissections of the temporal bone are an essential part of the training of ENT residents who aspire to perform middle ear surgery. These dissections provide surgeons with hands-on experience of the anatomy of the middle ear, which is crucial for developing surgical skills and confidence. Temporal bone dissection courses also allow surgeons to practice surgical techniques in a safe and controlled environment. This is important because middle ear surgery can be delicate and complex, and small inaccuracies can compromise the outcomes of such procedures. Studies consistently report FN injury (0.5% to 10%), dural tears (1% to 5%), and vascular injuries (1% to 3%) as potential complications associated with temporal bone surgery. These complications can not only compromise the intended surgical outcome but also pose a threat to patient safety [5,6,7]. Therefore, to perform these procedures safely and effectively, surgeons must have strong training and a wide knowledge of friendly landmarks before performing operations on live patients [1,8,9]. This study aimed to identify common mistakes and critical passages during the steps of temporal bone dissection among residents.

## 2. Materials and Methods

In total, 100 residents participated in this retrospective survey. The study exclusively recruited ENT residents across all four residency years. Participants were divided by year of residency, with equal numbers (20) recruited from the first two years, and a higher number (30) recruited from the last two years. All participants were under 32 years of age. The sample comprised 53 males and 47 females. Residents belonging to other specialties and practicing ENT specialists were excluded from the study. Participants were recruited from academic medical centers in Italy. After providing their year of residency and the number of temporal bone dissection courses attended during training years, participants were asked to complete a survey that included questions about their experience with cadaveric temporal bone dissections. In particular, the survey asked participants to identify the most critical steps when performing their last temporal bone dissection, and to describe the most common errors they had made at each step and if it was worth implementing pre-operative CT scans in temporal bone dissection courses. The responses from the survey were analyzed using descriptive statistics. The frequency of each critical step and error was calculated. Critical steps and tips to avoid mistakes during these procedures are provided in the discussion section.

## 3. Results

The five most critical steps during temporal bone dissection, as identified by participants, are shown in Table 1. The most common errors associated with each step are also shown in Table 1.

Participants from the first two years of residency and with less than five temporal bone dissection courses attended during their training years were more likely to experience critical issues during the first three steps. On the other hand, participants belonging to the last two years and with more experience with cadaveric temporal bone dissections (more than five temporal bone dissection courses attended during their residency) were more likely to identify the five critical steps correctly, but still encountered challenges during the final two steps. Notably, difficulties and criticalities were particularly pronounced when performing posterior tympanotomy. All participants agreed on the need to implement pre-dissection CT scans to reduce the rate of mistakes for each step.

## 4. Discussion

The results of this study suggest that cadaveric temporal bone dissections are an important part of the training of otolaryngology residents. These dissections provide surgeons with the opportunity to develop the skills and knowledge necessary to perform CM and PT safely and effectively [1,8,9]. The findings of this study also suggest that experience with cadaveric temporal bone dissections is associated with a lower risk of making errors during CM and PT. This implies that ENT residents should be encouraged to participate in as many cadaveric dissections as possible. However, it is important to analyze and discuss the critical steps to prevent mistakes that can be very serious in real patients. Despite advancements in surgical techniques, injuries to critical structures, including the SS, the dura of the MCF, and the FN continue to be documented in the literature. Therefore, attaining impeccable anatomical knowledge of the temporal bone remains paramount for aspiring oto-surgeons [5,6,7].

### 4.1. How to Avoid Mistakes in Step 1: Creation of the Mastoid Cavity

Since there are no important structures in the cortex, the surgeon should start the dissection with the largest available burr and the largest available suction-irrigator. Before commencing bone drilling, the bone surface must be irrigated with a suction-irrigator. A straight cut is then created along the temporal line posteriorly to the sinodural angle, while a second cut, immediately posterior to the posterior canal wall, is made perpendicular to the first and runs toward the mastoid tip. These two cuts define Macewen’s suprameatal triangle (Figure 1). The creation of a properly defined attack triangle with a large cutting burr is a critical initial step, enabling the subsequent dissection passages to proceed smoothly [10,11,12]. Inadequate triangle formation, which was found in first-year ENT residents, can significantly compromise subsequent dissection steps, potentially leading to difficulty in locating crucial landmarks.

Next, wide cortical removal and thinning of the posterior canal wall should be achieved before penetrating the antrum [9,10]. The posterior canal wall should be thinned to the point that the shadow of an instrument can be seen through the bone when the canal skin is elevated (Figure 2). 

### 4.2. How to Avoid Mistakes in Step 2: MCF Dura Identification

Identification of the dura of the MCF is of primary importance. Usually, this structure lies superior to the temporal line, and this is typically the case in pneumatized mastoids [13,14]. On the other hand, a low-lying dura is rarer and can be associated with poorly pneumatized mastoids. In these cases, the low tegmen mastoid plate can render the identification of the antrum more difficult [13,14]. Failure to locate the middle fossa plate leads to the insufficient removal of cells in the superior aspect of the dissection, limiting the exposure of the epitympanic area [13,14]. According to this survey, lacerating the dura was one of the most common mistakes that occurred during dissection. However, the participants who lacerated the dura of the MCF (mostly second-year residents) explained that this occurred in an attempt to obtain optimal exposure of the epytimpanic area for anatomical purposes (Figure 3), and even because sometimes they had to deal with non-pneumatized mastoids and low-lying tegmen plates.

The best way to expose the bony layer of the MCF dura is to burr the mastoid cortex along the temporal line until a change in bone color and sound is noted. This indicates the proximity of the dura of the MCF itself. The dissection must then proceed parallel to the curve that the dura normally forms toward the depth of the antrum. In the posterior aspect of the mastoid process, the temporal line seems to be a less accurate landmark with which to identify the dura. On the other hand, for working posteriorly, the parietal notch has been described as a more precise landmark. Therefore, to achieve safe MCF dura identification, it is worth keeping in mind the temporal line, in the anterior aspect of the dissection, and the parietal notch, in the posterior one. The exposure of the bony layer of the MCF is an extremely important step for optimal access to the sinodural angle [13,15,16,17,18] (Figure 4).

### 4.3. How to Avoid Mistakes in Step 3: Identification of the Sigmoid Sinus (SS)

Injury to the SS emerged as a critical step from this survey. Half of the participants who injured this structure attributed their error to an attempt to skeletonize it along its entire length, suggesting that such an event would be avoidable in a clinical setting. The remaining participants, who were younger and had fewer than five dissection courses conducted in their life, reported that their injury occurred due to attempting to identify the SS before identifying the dura of the MCF. A consistent finding among participants who injured the SS was that non-pneumatized mastoids, which hindered dissection, were a constant cause of injury. Hence, even if steps are performed in the correct order, injury to this structure can still occur in non-pneumatized mastoids. This is because the SS is harder to visualize, the sinodural angle is narrower in non-pneumatized bone, and, as a result, surgeons may inadvertently injure the sinus during dissection [19]. This finding could apply to the clinical setting, as a poorly pneumatized mastoid is often found in many surgical procedures for inflammatory middle ear pathology.

On the other hand, well-pneumatized mastoids are more typical of neuro-otological procedures. Therefore, remembering that the SS is typically located a few millimeters below the mastoid cortex, the following are important considerations for avoiding injury to the SS: Firstly, it is essential to avoid immediate identification of the SS. Instead, it is appropriate to first identify the dura of the MCF (Figure 4). The surgeon must be very careful of changes in sound and color and must take extra caution in poorly pneumatized mastoids, as the sinus is less visible and the sinodural angle is narrower [17,18,19].

### 4.4. How to Avoid Mistakes in Step 4: Opening the Mastoid Antrum

Together with the PT, the opening of the mastoid antrum (Figure 5) emerged as a critical and error-prone step in middle ear surgery, both for third and fourth-year residents. The most common mistakes that occurred during mastoid antrum opening included the tearing of the dura of the MCF, injury of the ossicular chain, and injury of the FN. Once again, the occurrence of errors during mastoid antrum opening was significantly higher in non-pneumatized mastoids, likely due to the reduced dimensions of the antrum and the lower position of the tegmen plate [14,20]. When asked about other reasons why they had problems opening the mastoid antrum, participants reported that errors were also due to other factors such as failure to identify previous surgical landmarks, inadequate thinning of the posterior canal wall, and inability to visualize the lateral semicircular canal, which led to injury of the seventh cranial nerve. According to the participants and their tutors during the dissection courses, it is likely that the inability to correctly identify the lateral semicircular canal was due to the fact that participants often thought they had reached it. Still, it was the Koerner septum, a non-pneumatized structure that contributes to attic blockage and the development of middle ear pathology [21,22,23]. This structure is by definition more superficial than the compact bone of the lateral semicircular canal [21,22,23]. Being a non-pneumatized structure, it is easily understandable why this identification is even more difficult in the case of poorly pneumatized mastoids. [21,22,23]. Therefore, the mastoid antrum should be identified by first locating the Koerner septum, when present, and then perforating it in the anterosuperior quadrant of the dissection. A progressive deepening of the dissection, with careful thinning of the posterior canal wall of the canal and atraumatic contact with the dura of the MCF, allows the identification of the mastoid antrum. The fundamental landmarks, such as the labyrinthine compact bone of the lateral semicircular canal, are found at the base of the antrum [24]. The identification of the lateral semicircular canal allows for the exposure of the incus fossa, the epitympanum antero-superiorly, and the second genu of the FN infero-medially [1,2,17,18,24]. The anatomy of the antrum changes in accordance with the type of middle ear surgery that the surgeon is facing: in CI, the anatomy is usually well preserve; on the other hand, when performing inflammatory pathologies of the middle ear such as cholesteatoma, mastoid inflammatory changes and eroded incus are found in more than 90% of cases [2,25,26].

### 4.5. How to Avoid Mistakes in Step 5: Posterior Tympanotomy (PT)

Facial recess identification (Figure 6) and PT (Figure 7) were identified as the most critical steps in this survey, even for senior residents. Junior ENT residents did not report any criticalities in this step, simply because they were unable to reach this point in their first dissections. Having had difficulties in the first three steps, it was therefore impossible to proceed further. Participants reported the following errors: CT injury, FN injury, tympanic membrane perforation, posterior canal wall violation, and incus buttress rupture. Even in this last step, there was accordance among participants who claimed that a not well-pneumatized mastoid increased the risk of mistakes during a PT. This fact can be explained by the narrower space for PT and the more laterally running facial nerve in poorly pneumatized mastoids of patients suffering chronic inflammatory diseases of the middle ear [27,28]. According to the participants, errors during the identification of the facial recess and the execution of PT can be attributed to a variety of causes not only related to the more complex anatomy of their dissection. Imperfect identification of the previous landmarks affected all errors in this step. Instead, for participants who had correctly identified the previous landmarks, the errors were attributed to the following: (1) use of incorrect-sized drills during PT with damage to the CT, the FN, and the incus buttress; (2) excessive thinning of the posterior canal wall, which caused an interruption; (3) damage to the tympanic membrane, which was attributed to incorrect visualization during dissection maneuvers, the accidental section of the CT, and therefore to the injury of a landmark that could lead to the membrane itself. To perform a correct PT, it is necessary to identify the landmarks of the facial recess (Figure 6, Figure 7 and Figure 8). These landmarks are represented by the external genu of the FN medially, the incus buttress superiorly, the CT laterally, and the tympanic membrane anterolaterally [17,18,28]. Normally, the incus points to the facial recess like an arrow [29] (Figure 6, Figure 7 and Figure 8). In about 43% of cases, it is seen as a big cell that heralds the entry of the middle ear, and this is why this cell is known as a “herald cell”. This cell can be a reliable marker for the identification of the facial recess [1] (Figure 7).

The dissection of the facial recess begins by identifying the descending portion of the FN in the mastoid. It is essential to use copious irrigation to constantly visualize the bone that is being drilled and monitor for any color changes. Generally, the dissection is performed with a cutting burr until a color change is identified and then with a diamond burr. Since bone color changes occur before the facial sheath is exposed, it is possible to preserve a thin layer of bone over the nerve, avoiding damage to its sheath. It is often possible to identify the cells of the facial recess by thinning the posterior wall of the canal (Figure 6 and Figure 7). To complete the dissection of the facial recess, it is necessary to use smaller burs as the recess is rarely larger than 3 millimeters. Inferiorly, it is necessary to identify the CT as it emerges from the FN [9,15,17,18,30] (Figure 8). Once the PT is complete, the tympanic cavity can be visualized through the facial recess (Figure 9). The round window can be easily identified inferior to the stapes landmarks. Superiorly, the incus buttress should be preserved to prevent the disinsertion of the ligaments of the short process of the incus, and thus avoid the possible dislocation of the incus itself. However, it is worth mentioning that sometimes the visibility of the round window may not be optimal due to factors that can influence its visibility, such as narrower facial recesses and unfavorable cochlear rotation (Figure 10 and Figure 11) [9,15,17,18,30]. Several authors have found a narrower facial recess as well as a more laterally running course of the FN, and therefore a higher risk of damaging it in patients with middle ear inflammation [27,28].

This study suggests that the last two steps are the most important to focus on, as they involve mostly ENT senior residents who will soon be approaching surgery on living patients. These are two essential steps for the treatment of numerous ear pathologies [2,3,4,27,31]. The facial recess is often a pathway for the spread of middle ear pathology from the tympanic cavity to the mastoid. The opening of the facial recess is of great importance in any chronic ear pathology to create an additional pathway for ventilation to the mastoid. This technique also allows for better visualization of the tympanic cavity in chronic otitis media and permits exposure of the horizontal portion of the FN during the decompression of the nerve itself. Not only do chronic middle ear diseases and some tumors arising in the middle ear cavity, but many more procedures and pathologies depend on these steps [2,3,27,31]. In fact, it is also the pathway to access the round window through which to insert the electrode in CI surgery [2,26]. This is very important because nowadays, there is an increase in the average age of the general population and therefore also of patients who require a rehabilitative surgical treatment such as cochlear implants. Not only does it seem that these can rehabilitate hearing in the elderly, improving their quality of life, but it seems that their role could be a potent weapon, as through rehabilitation and hearing recovery, the patient’s isolation could be prevented and therefore diseases such as dementia could be slowed down [32,33]. While further studies are warranted, the potential benefits of early acoustic rehabilitation in adults have significant implications for the field of otology. If future research confirms these benefits, the surgical steps discussed are likely to have even greater importance and become more widely adopted by otologists around the world, as they effectively address the needs of an aging population. For these reasons, it is necessary that young people who aspire to become ear surgeons, already in the early years of their residency, take dissection courses on the temporal bone with a careful pre-dissection study of the surgical steps to perform. To improve the training of young oto-surgeons, all participants agreed on the implementation of a preoperative CT scan of the temporal bone in these dissection courses. This would be very useful and appropriate because it can help young surgeons learn how to interpret radiological findings before an operation and raise awareness on how to identify surgical pitfalls such as high and dehiscent jugular bulb, exposed carotid artery in the tympanic cavity, low middle cranial fossa dura, dehiscences in the tegmen tympani, hypo-pneumatized mastoid, FN course, Koerner septum, and other anomalies [15,21,22,23,24,28].

## 5. Conclusions

Thorough training in temporal bone dissection is essential for surgeons who intend to perform ear surgery on living patients. A meticulous pre-dissection study, manual dexterity, experience, and the disciplined and systematic identification of surgical landmarks are crucial for successful dissection and the avoidance of complications in real-world patients. Pre-dissection CT scans may represent a very useful tool to improve the training of ENT residents and might be considered for future studies. 

## 6. Limitations of the Study

The principal limitation of this study is its retrospective design. Participants’ recollection of the number of temporal bone dissection courses attended and the specific challenges encountered during dissection may be susceptible to recall bias.

## Figures and Tables

**Figure 1 jpm-14-00349-f001:**
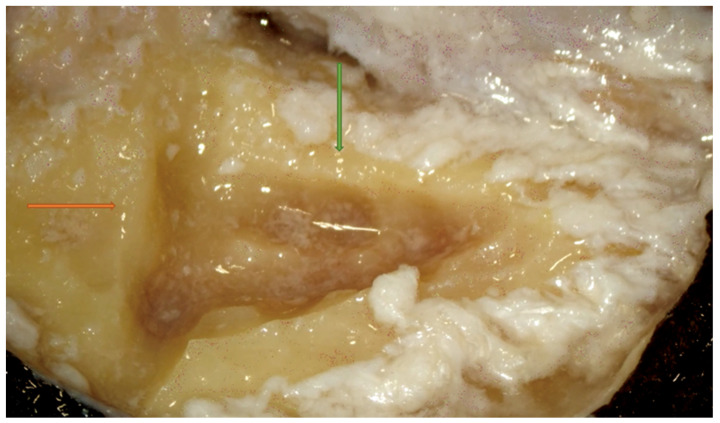
Right side. Macewen’s suprameatal triangle. A first straight cut (orange arrow) is made along the temporal line, while the second cut (green arrow) is made perpendicular to the first and immediately posterior to the posterior canal wall.

**Figure 2 jpm-14-00349-f002:**
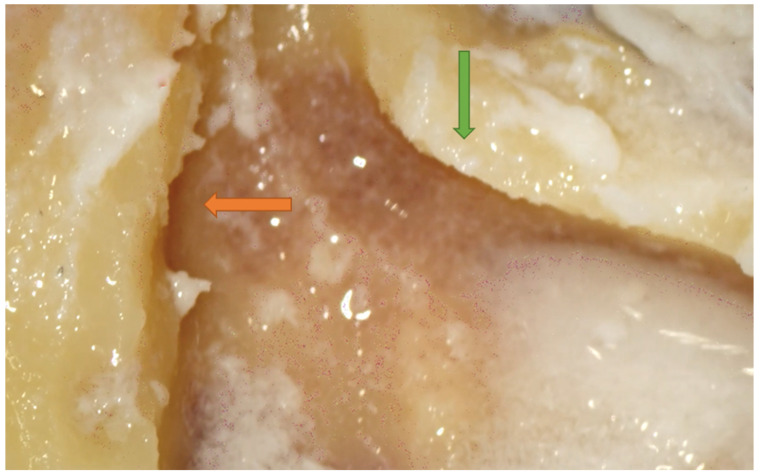
Right side. Thinning of the posterior canal wall (green arrow), and MCF dura identification (orange arrow).

**Figure 3 jpm-14-00349-f003:**
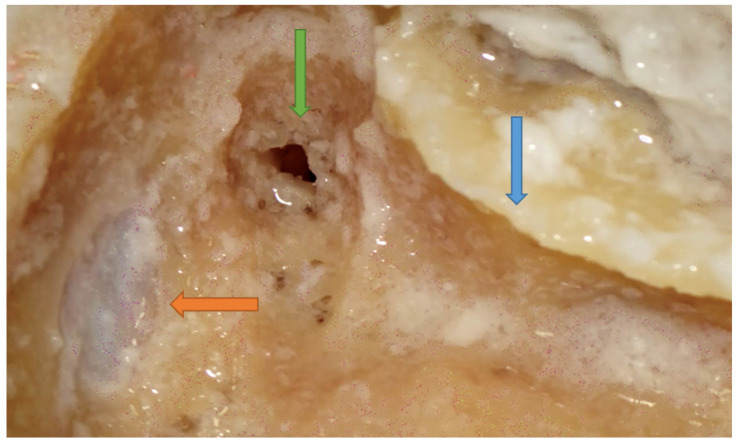
Right side. Dura uncovering (orange arrow) after resident attempted to achieve wide exposure of the epitympanic area. Note that the antrum is about to be opened (green arrow). Note the thinned posterior canal wall (blue arrow).

**Figure 4 jpm-14-00349-f004:**
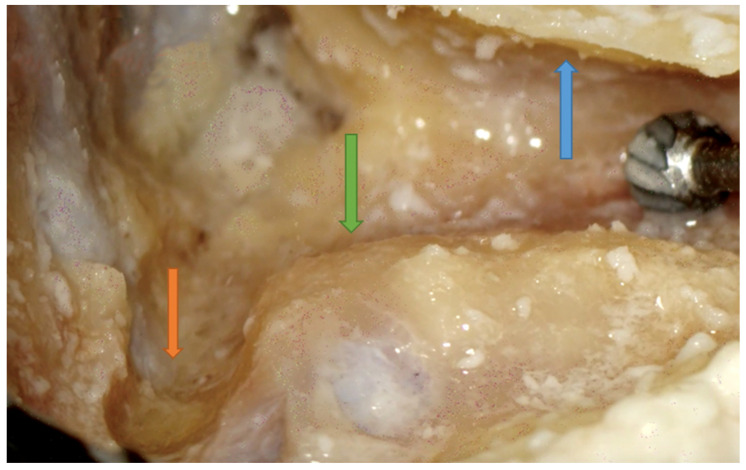
Right side. Focus on the sinodural angle (orange arrow) and SS (green arrow). The SS must be identified after the dura of the MCF. Blue arrow showing a thinned posterior canal wall.

**Figure 5 jpm-14-00349-f005:**
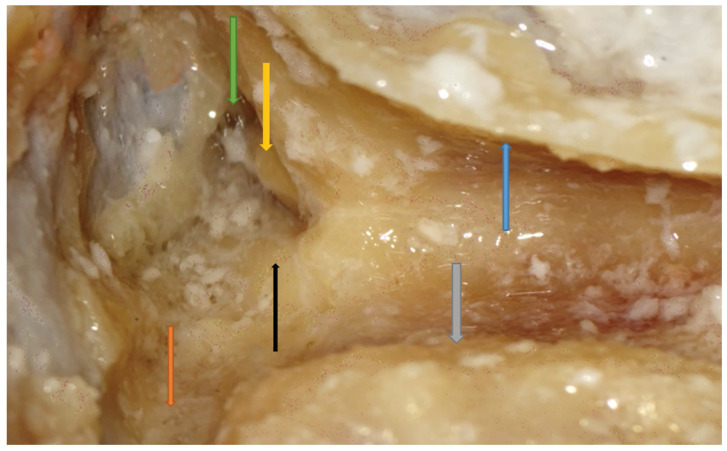
Right side. Opened mastoid antrum (green arrow) and incus visualization (yellow arrow). The salience of the lateral semicircular canal is evident (black arrow). The sinodural angle (orange arrow) and SS (grey arrow) are shown. The dura of the MCF is widely exposed for dissection purposes. Other important steps to reach this point include keeping a very thin posterior canal wall (blue arrow) and perforating the Koerner septum.

**Figure 6 jpm-14-00349-f006:**
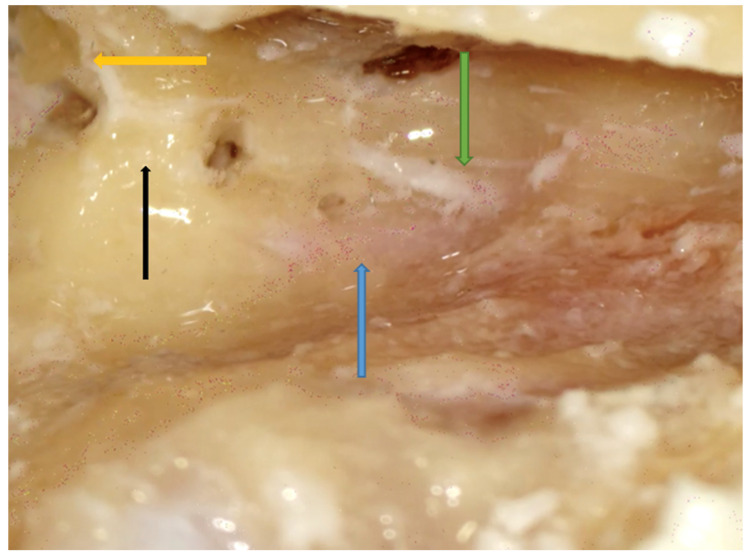
Right side. Facial recess landmark identification. Incus buttress (black arrow), FN (blue arrow), and CT (green arrow) are shown. Note that the incus (yellow arrow) points right to the facial recess.

**Figure 7 jpm-14-00349-f007:**
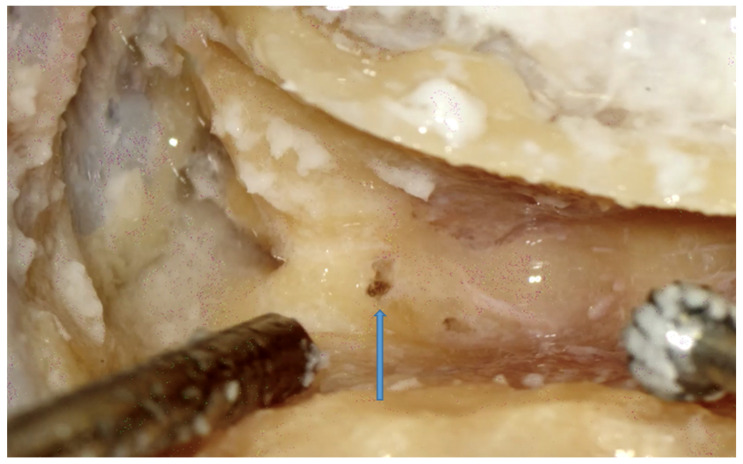
Facial recess. A small herald cell (blue arrow) is present.

**Figure 8 jpm-14-00349-f008:**
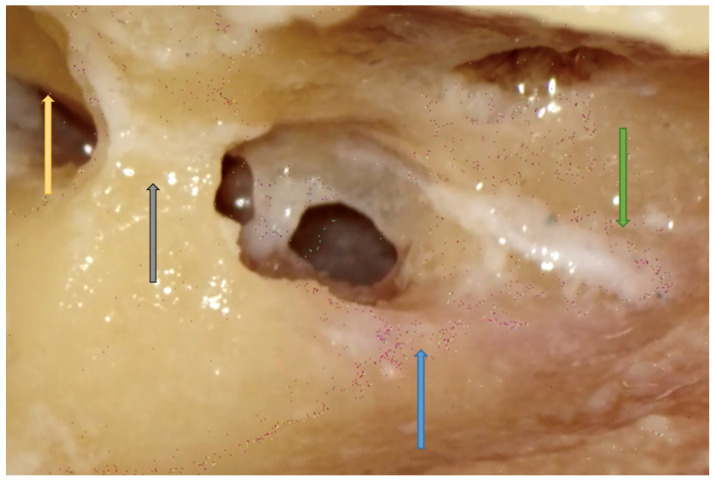
Right side. Opening the facial recess through PT. Note the CT (green arrow) as it emerges from the FN (blue arrow). The incus (yellow arrow) and the incus buttress (grey arrow) are pointed out.

**Figure 9 jpm-14-00349-f009:**
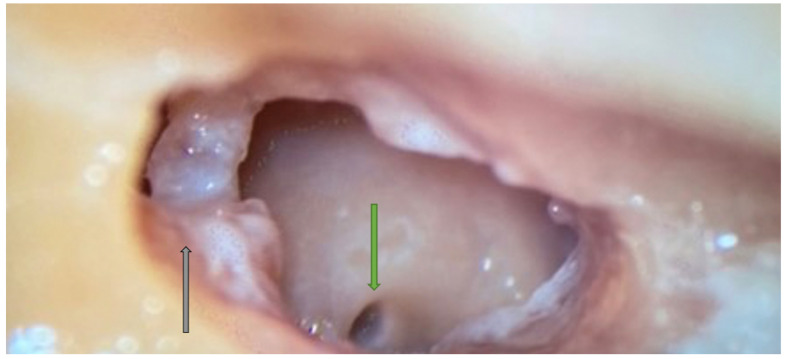
Right side. Tympanic cavity through the facial recess. Note that the round window (green arrow) is located inferior to the stapedial landmarks (grey arrow).

**Figure 10 jpm-14-00349-f010:**
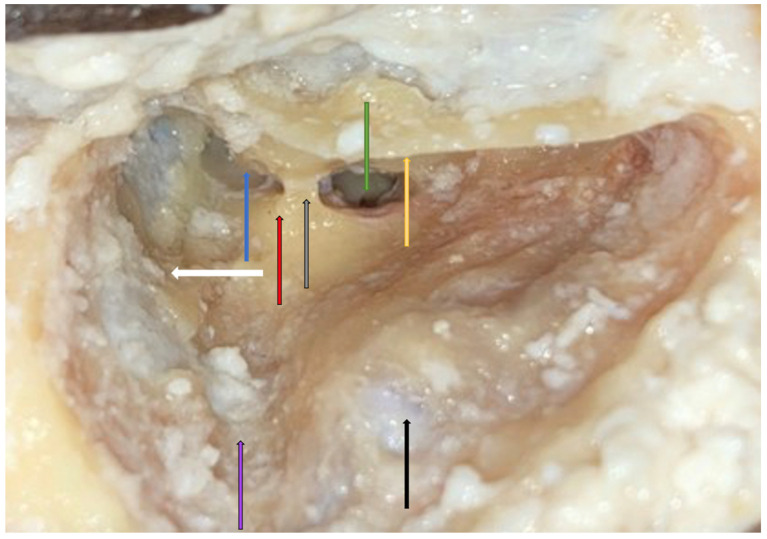
Right side. Facial recess with good visibility of the round window (green arrow) even without a microscope. All superficial and deep landmarks are shown: the dura of the MCF (white arrow), thinned posterior canal wall (yellow arrow), sigmoid sinus (black arrow), sinodural angle (purple arrow), lateral semicircular canal (red arrow), incus (blue arrow), and incus buttress (grey arrow).

**Figure 11 jpm-14-00349-f011:**
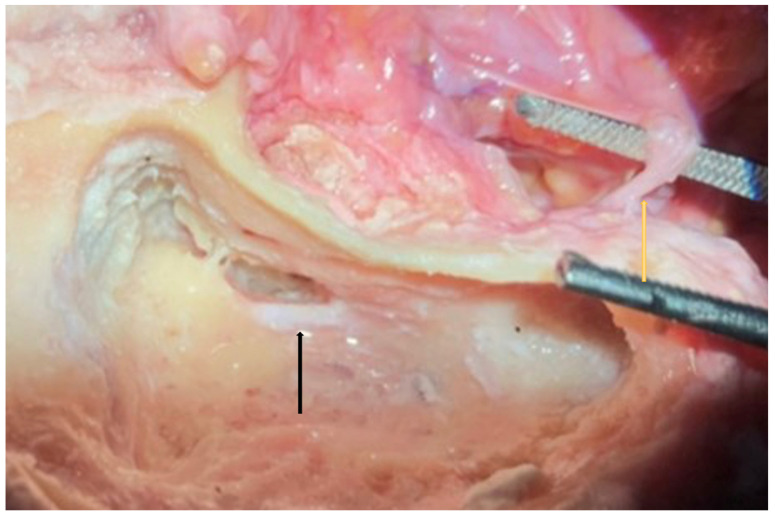
Right side. In this temporal bone dissection, the visibility of the round window was harder due to the smaller size of the facial recess and the unfavorable cochlear rotation. This resident performed parotidectomy the day before, and therefore, it is worth noticing the mastoid portion of the FN (black arrow) and its exit from the stilomastoid foramen (yellow arrow).

**Table 1 jpm-14-00349-t001:** Critical steps and errors during temporal bone dissection.

Step	Criticality	Error
1. Creation of the mastoidectomy cavity with thinning of the posterior canal wall	High	Poor exposure of the mastoid cavity, making subsequent steps more difficult
2. Identification of the middle cranial fossa dura	High	Lacerations of the middle cranial fossa dura
3. Identification of the sigmoid sinus	High	Lacerations of the sigmoid sinus
4. Opening of the mastoid antrum	High	Lacerations of the middle cranial fossa dura, damage to the ossicular chain, facial nerve injury
5. Performing the posterior tympanotomy	High	Facial nerve injury, chorda tympani nerve injury, tympanic membrane perforation, posterior canal wall disruption, incus wall rupture

## Data Availability

Data are available upon reasonable request.

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
