# Peer review of "Critical Steps and Common Mistakes during Temporal Bone Dissection: A Survey among Residents and a Step-by-Step Guide Analysis"

_jpm, 2024, doi:10.3390/jpm14040349_

Round 1

Reviewer 1 Report

Comments and Suggestions for Authors

• Title and abstract:
o The title is too long.
o The abstract also represents the study well.
o The grammar in this section should be reviewed.
o The keywords are properly selected.

• Introduction:

Line 54 Correct the sentence The study aimed to identify This study aimed to identify, common mistakes and critical passages during the steps of temporal bone dissection among ENT residents.

I recommend further literature review about the incidence and percentage of complications of Temporal bone surgeries, risk factors.
o The Introduction is explained nicely.
o The study was justified.
o The rationale is presented well, and the objectives are clear.
o The grammar in this section should be reviewed.

• Method

The study design is not mentioned in this section

Sample Size Sampling Technique

No inclusion or exclusion criteria

Data Collection Methods (online survey)
• Discussion and Conclusions

Very little literature is discussed and lacks a detailed description of the influencing factors and the argument.

Report any limitation observed by the authors, such as retrospective design and its related biases.

Reference (29) is about Hearing Aids and Cochlear Implants in the Prevention of Cognitive Decline and 384 Dementia-Breaking Through the Silence, clarify the relation to your study

Reference (30) is about  cochlear implantation  and prevention of cognitive decline   long-term follow-up, , clarify the relation to your study.

Findings were summarized and compared to other studies

Significance of this study was explained well

Comments on the Quality of English Language

Good quality

Author Response

Dear Reviewer,

I express my gratitude for your insightful comments and constructive suggestions aimed at elevating the calibre of our manuscript.

  1. As per previous recommendations, the title has been shortened and the abstract has undergone grammatical review with the assistance of a native English speaker
  2. In accordance with your suggestions, the previously mentioned sentence in the introduction has been corrected as you indicated. Additionally, the introduction has been enriched with further literature and scientific articles encompassing complication rates and associated risks in temporal bone surgery.
  3. Within the methodology section, the study design and the sampling technique have been clarified. Moreover, the criteria for participant inclusion and exclusion have been outlined
  4. The limitations of the study have been highlighted
  5. Clarifying reference 29 e 30: The surgical approaches described in this paper hold substantial importance, as they enable access to the round window area during cochlear implantation surgery. As highlighted in references 29 and 30, cochlear implants may play a vital role in addressing the needs of an increasingly aging population. With the rise in the average age of the population, ensuring a good quality of life for all patients becomes increasingly significant. Cochlear implants have gained widespread recognition for their ability to restore hearing, avoid isolation and improve communication in individuals with severe hearing loss. Current researches explore the potential of cochlear implantation in adults to prevent or slow down cognitive decline. While further studies are warranted, the potential benefits of early acoustic rehabilitation in adults have significant implications for the field of otology. If future research confirms these benefits, the surgical passages discussed in this paper are likely to gain even greater importance and become more widely adopted by otologists, as they effectively address the needs of an aging population

Best regards,

Giovanni Motta

Reviewer 2 Report

Comments and Suggestions for Authors

       The authors present a comprehensive guide to the mastoidectomy technique, which will be of particular interest to the medical community, especially ENT specialists and residents. The paper offers extensive details in a structured and coherent way. I have some suggestions for improving the manuscript form as follows:

1. I recommend adding arrows to the images to better visualize the structures and increase clarity.

2. I noticed that the methodology and results sections of the report appear to be incomplete. I recommend expanding these sections, particularly the methodology section. It would be helpful to include details such as the study type and time, inclusion criteria for the participants, and information about gender distribution, age, and year of residency. Also, the type of questionnaire used and the questions addressed to the residents need to be considered.

Author Response

Dear Reviewer,

I express my gratitude for your insightful comments and constructive suggestions aimed at elevating the calibre of our manuscript.

  1. As you suggested, arrows have been added to images to improve the clarity of this paper
  2. The methodology and results section has been expanded. In particular, the methodology section has been expanded to provide a more detailed and comprehensive overview of the study design and data collection procedures. The participant inclusion and exclusion criteria have been specified. Moreover, information regarding gender distribution, year of residency and the questions posed to the participants were specified.

I hope that the enhanced level of details strengthens the methodological rigor of the study and provides a clearer picture of our paper.

Best regards,

Giovanni Motta